# Multi-Source Neural Model for Machine Translation of Agglutinative Language

**Yirong Pan** [1,2,3], **Xiao Li** [1,2,3,*], **Yating Yang** [1,2,3,*] and **Rui Dong** [1,2,3]

[1] Xinjiang Technical Institute of Physics & Chemistry, Chinese Academy of Sciences, Urumqi 830011, China; panyirong15@mails.ucas.ac.cn (Y.P.); dongrui@ms.xjb.ac.cn (R.D.)

[2] Department of Computer and Control, University of Chinese Academy of Sciences, Beijing 100049, China

[3] Xinjiang Laboratory of Minority Speech and Language Information Processing, Urumqi 830011, China

[*] Correspondence: xiaoli@ms.xjb.ac.cn (X.L.); yangyt@ms.xjb.ac.cn (Y.Y.)

**Abstract:** Benefitting from the rapid development of artificial intelligence (AI) and deep learning, the machine translation task based on neural networks has achieved impressive performance in many high-resource language pairs. However, the neural machine translation (NMT) models still struggle in the translation task on agglutinative languages with complex morphology and limited resources. Inspired by the finding that utilizing the source-side linguistic knowledge can further improve the NMT performance, we propose a multi-source neural model that employs two separate encoders to encode the source word sequence and the linguistic feature sequences. Compared with the standard NMT model, we utilize an additional encoder to incorporate the linguistic features of lemma, part-of-speech (POS) tag, and morphological tag by extending the input embedding layer of the encoder. Moreover, we use a serial combination method to integrate the conditional information from the encoders with the outputs of the decoder, which aims to enhance the neural model to learn a high-quality context representation of the source sentence. Experimental results show that our approach is effective for the agglutinative language translation, which achieves the highest improvements of +2.4 BLEU points on Turkish–English translation task and +0.6 BLEU points on Uyghur–Chinese translation task.

**Keywords:** artificial intelligence; neural machine translation; agglutinative language translation; complex morphology; linguistic knowledge

## 1. Introduction

With the rapid development of artificial intelligence and deep learning, neural networks are widely applied to various fields ranging from computer vision [1,2], speech recognition [3,4], and natural language processing (NLP) [5–8]. The standard neural machine translation (NMT) model [9–12] employs the encoder to map the source sentence into a continuous representation vector, then it feeds the resulting vector to the decoder to generate the target sentence, which directly learns the translation relationship between two distinct languages from the bilingual parallel sentence pairs. Recently, by exploiting advanced neural networks, such as long short-term memory (LSTM) [13], gate recurrent unit (GRU) [14], and attention mechanism [15], NMT has become the current dominant machine translation approach, and it achieves impressive performance on many high-resource language pairs, such as Chinese–English translation and English–German translation.

However, existing NMT models still struggle in the translation task of agglutinative languages with complex morphology and limited resources, such as Turkish to English and Uyghur to Chinese. The morpheme structure of the word in agglutinative language is formed by a stem followed by a sequence of suffixes (since the words only have a few prefixes, we simply combine the prefixes with a

stem into the stem unit), which can be denoted as: word = stem + suffix1 + suffix2 + ... + suffixN [16]. For example, in the Turkish phrase "küçük fagernes kasabasındayım" (I'm in a small town of fagernes), the morpheme structure of the word "kasabasındayım" (I'm in town) is: kasaba + sı + nda + yım. Due to the fact that the suffixes have many inflected and morphological variants depending on the case, tense, number, gender, etc., the vocabulary size of an agglutinative language is considerable even in small-scale training data. Moreover, a word can express the meaning of a phrase or sentence. Thus, there are many rare and out-of-vocabulary (OOV) words in the training process, which leads to many inaccurate translation results [17] and increases the NMT model complexity.

Recently, researchers attempted to explicitly use the source-side linguistic knowledge to further improve the NMT performance. Sennrich and Haddow generalized the word-embedding layer of the encoder to accommodate for additional linguistic input features including lemma, sub-word tag, part-of-speech (POS) tag, dependency label, and morphological tag for German–English translation [18]. Eriguchi et al. proposed a tree-to-sequence-based model for English–Japanese translation, which encodes each phrase in the source parse tree and used the attention mechanism to align both the input words and phrases with the output words [19]. Yang et al. improved the above work by encoding each node in the source parse tree with the local and global context information, and they utilized a weighted variant of the attention mechanism to adjust the proportion of the conditional information for English–German translation [20]. Li et al. combined the source-side sentence with its linearized syntactic structure, which makes the NMT model automatically learn useful language information for Chinese–English translation [21]. Currey and Heafield modified the multi-source technique [22] for English–German translation, and they exploited the syntax structure of the source-side sentence by employing an additional encoder to encode the linearized parse tree [23]. Li et al. presented a linguistic knowledge-aware neural model for both English–Chinese translation and English–German translation, which uses a knowledge gate and an attention gate to control the information from the source words and the linguistic features of POS tag, named entity (NE) tag, chunk tag and dependency label [24].

However, the above works mostly pay attention to the high-resource machine translation tasks with large-scale parallel data and sufficient semantic analysis tools, which lacks the consideration of agglutinative language translation with complex morphology and limited resources. In this paper, we propose a multi-source neural model for the machine translation task on agglutinative language. We consider that enhancing the ability of the NMT model in capturing the semantic information of the source-side sentence is beneficial to compensate for both the corpus scarcity and data sparseness. The followings are our main contributions:

- Focusing on the complex morphology of the agglutinative language, in contrast to the standard NMT model that uses a single encoder, we utilize a multi-source NMT framework consisting of a word-based encoder and a knowledge-based encoder to encode the word feature and the linguistic features, respectively, which aims to incorporate the source-side linguistic knowledge into the NMT model.
- For the purpose of enriching each source word's representation in the NMT model, we extend the input embedding layer of the knowledge-based encoder to allow for the word-level linguistic features of lemma, POS tag and morphological tag.
- In the consideration of enhancing the NMT representation ability on the source-side sentence, we use a serial combination method to hierarchically combine the conditional information from the encoders, which helps to learn a high-quality context representation. Firstly, the representation of the source-side linguistic features integrates with the representation of the target sequence. Secondly, the resulting vector integrates with the representation of the source sequence to generate a context vector. Finally, the context vector is employed to predict the target word sequence.

Experimental results in Turkish–English and Uyghur–Chinese machine translation tasks show that the proposed approach can effectively improve the translation performance of the agglutinative

language, which indicates the validity of the multi-source neural model on using the source-side linguistic knowledge for morphologically rich languages.

## 2. Related Works

Recently, many researchers showed their great interest in improving the NMT performance in the low-resource and morphologically-rich machine translation tasks. The first line of research attempted to utilize external monolingual data [25,26]. Gulcehre et al. presented an effective way to integrate a language model that trained on the target-side monolingual data into the NMT model [27]. Sennrich et al. paired the target-side monolingual data with automatic back-translation and treated it as additional parallel data to train the standard NMT model [28]. Ramachandran et al. employed an unsupervised learning method that first uses both the source-side and target-side monolingual data to train two language models to initialize the encoder and decoder, then fine-tunes the trained NMT model with the labelled dataset [29]. Currey et al. utilized the target-side monolingual data and copied it to the source-side as additional training data, then mixed with the original training data to train the NMT [30]. The second line of research attempts to leverage other languages for the zero-resource NMT [31,32]. Cheng et al. proposed a pivot-based method that first translates the source language into a pivot language, then translates the pivot language into the target-side language [33]. Zoph et al. utilized a transfer learning method that first trains a parent model on the high-resource language pair, then transfers the learned parameters to initialize the child model on the low-resource language pair [34].

The multi-source neural model was first used by Zoph and Knight for multilingual translation [22]. It can be seen as a many-to-one setting in the multi-task sequence-to-sequence (se2seq) learning [35]. The model consists of multiple encoders with one encoder per source language and a decoder to predict the required target language, which is effective to share the encoder parameters and enhance the model representation ability. Multi-source neural models are widely applied to the fields of machine translation [36,37], automatic post-editing (APE) [38,39] and semantic parsing [40,41].

NLP tasks are performed by using supervised learning with large-scale labelled training data. However, since the artificial labelled data is limited, it is valuable to utilize the additional resources to further improve the model performance. In recent years, many basic NLP tasks such as POS tagging, named entity recognition, and dependency parsing are used as prior knowledge to improve the higher-level NLP tasks such as summarization, natural language inference and machine translation [42–45]. Generally speaking, the usage of linguistic annotations is helpful to better identify the word in the context. Our approach follows this line of research.

## 3. Materials and Methods

### 3.1. Standard NMT Model

In this paper, we followed the NMT model proposed by Vaswani et al. [12], which was implemented as a single-source Transformer model with an encoder–decoder framework as shown in Figure 1. We employed the basic model as our baseline. We will briefly summarize it in this section.

Given the word sequence $x = (x_1, \ldots, x_m)$, both the input and output embedding layers map it into a word-embedding matrix $e = (e_1, \ldots, e_m)$, where $e_i$ is computed by

$$e_i = x_i \cdot E_x, \tag{1}$$

where $x_i \in \mathbb{R}^{1 \times K_x}$ is the one-hot vector, $E_x \in \mathbb{R}^{K_x \times d_x}$ is the word-embedding matrix, $K_x$ is the vocabulary size and $d_x$ is the word-embedding size. In order to make use of the word order in the sequence, Transformer provides the "positional encoding" function for the embedding layers, which uses sine and cosine functions of different frequencies by

$$PE(pos, 2i) = sin\left(pos/10,000^{2i/d_x}\right), \tag{2}$$

$$PE(pos, 2i + 1) = cos\left(pos/10,000^{2i/d_x}\right),$$

where *pos* is the position of the word in the sequence and *i* is the dimension of the embedding matrix. The positional embedding matrix is the sum of the word-embedding matrix and the positional encoding matrix.

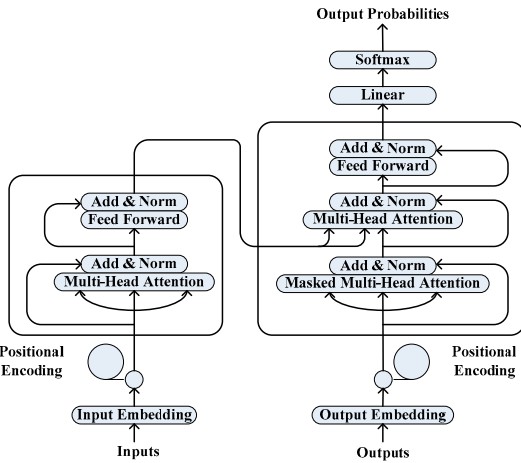

**Figure 1.** The structure of the single-source Transformer model with the encoder–decoder framework.

The encoder is composed of a stack of *N* identical layers and each layer has two sub-layers consisting of the multi-head self-attention and the fully connected feed-forward network. The multi-head self-attention maps the query set *Q*, the key set *K*, and the value set *V* into an attention matrix by

$$\text{MultiHead}(Q, K, V) = \text{Concat}(\text{head}_1, \ldots, \text{head}_\text{h})W^E, \tag{3}$$

$$\text{head}_\text{i} = \text{Attention}\left(QW_i^Q, KW_i^K, VW_i^V\right),$$

$$\text{Attention}(Q, K, V) = \text{softmax}\left(\frac{QK^\text{T}}{\sqrt{d_k}}\right)V,$$

where *Q*, *K*, and *V* are the vector sets of query, key, and value. The dimension of query, key, and value is $d_k$, $d_k$, $d_v$, respectively, and they are initialized by applying a linear transformation on the output of the previous sub-layer. Then, the fully connected feed-forward network performs the position-wise computation on the output *X* of the previous multi-head self-attention layer by

$$\text{FFN}(X) = \max\left(0, XW_1{}^E + b_1{}^E\right)W_2{}^E + b_2{}^E, \tag{4}$$

The decoder is also composed of a stack of *N* identical layers, where each layer has three sub-layers consisting of the multi-head self-attention, the multi-head attention and the fully connected feed-forward network. The multi-head self-attention was first applied on the target-side positional embedding matrix as the same with the encoder. Then, the multi-head attention builds an attention model between the encoder and the decoder, which utilizes the *K* and *V* vector sets from the outputs of the encoder and the *Q* vector set from the previous layer of the decoder as inputs to generate a context representation by

$$\text{MultiHead}(Q, K, V) = \text{Concat}(\text{head}_1, \ldots, \text{head}_\text{h})W^D, \tag{5}$$

Then, the fully connected feed-forward network is applied on the context representation *Y* by

$$\text{FFN}(Y) = \max\left(0, YW_1{}^D + b_1{}^D\right)W_2{}^D + b_2{}^D, \tag{6}$$

Finally, a linear network layer followed by a softmax layer was applied on the above generated vector to predict the target word sequence. In addition, the residual connection [46] was employed around each sub-layer followed by layer normalization [47].

### 3.2. Multi-Source Neural Model

In this paper, we proposed a multi-source neural model that employs two separate encoders consisting of the word-based encoder and the knowledge-based encoder to encode the source word features and the source-side linguistic features, respectively. Figure 2 shows the framework of our multi-source neural model. All the sub-layers in the model produced the output dimension of $d_x = 512$, and $d_k = d_v = d_x/h$. The following sections will describe the proposed model in detail.

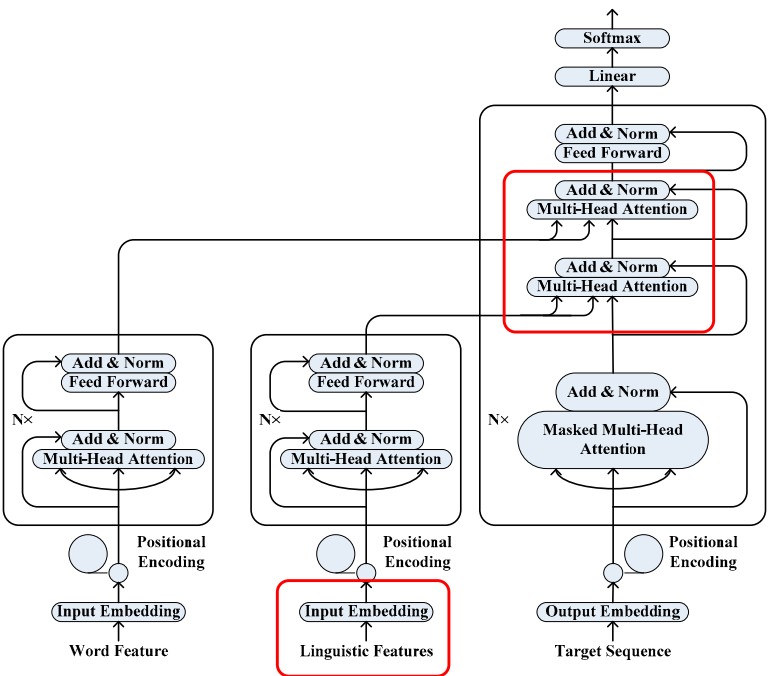

**Figure 2.** The framework of the proposed multi-source neural model.

### 3.2.1. Word-Based Encoder

The word-based encoder was employed to encode the source word features in the same way the encoder in the standard NMT model we previously mentioned does. The word-based encoder outputs the representation vector $H_1$ of the source sequence and the vector sets of $Q$, $K$, and $V$.

### 3.2.2. Knowledge-Based Encoder

In contrast to the encoder in the standard NMT model, the input word-embedding layer of the knowledge-based encoder was extended to incorporate each source word's linguistic features into the NMT model as shown in Figure 3.

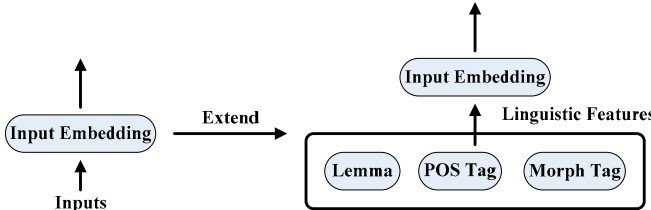

**Figure 3.** The input-embedding layer of the knowledge-based encoder.

Given the linguistic annotation sequences $k_1 = (k_{11}, \ldots, k_{1m})$, $\ldots$, and $k_F = (k_{F1}, \ldots, k_{Fm})$, the word-embedding layer maps them to a embedding matrix $e^* = (e_1^*, \ldots, e_m^*)$, where $e^*_i$ is computed by

$$e_i^* = \bigcup_{t=1}^{F} k_{ti} \cdot E_t, \tag{7}$$

where $\bigcup$ is the vector concatenation operator, $k_{ti} \in \mathbb{R}^{1 \times K_t}$ is the one-hot vector, $K_t$ is the vocabulary size of the $t$-th feature, $E_t \in \mathbb{R}^{K_t \times d_t}$ is the embedding matrix of the $t$-th feature, and $d_t$ is the embedding size of the $t$-th feature and $\sum_{t=1}^{F} d_t = d_x$. The knowledge-based encoder outputs the representation vector $H_2$ of the linguistic feature sequences and the vector sets of $Q^*$, $K^*$, and $V^*$.

### 3.2.3. Serial Combination Method

A simple combination method [48] is to concatenate the outputs of the encoders into a single vector, then apply a linear transformation and a non-linear transformation on the resulting vector by

$$h = tanh(\boldsymbol{W}_\theta [H_1; H_2]), \tag{8}$$

where $W_\theta$ is the trainable weight matrix. However, this method cannot effectively utilize the other parameters in the NMT model or dynamically control the proportions of the language information. Inspired by Libovick et al. [49], we utilized a serial combination method to integrate the conditional information from the encoders with the outputs of the decoder, which additionally inserted a second multi-head attention layer in the decoder to perform the attention function.

Firstly, the multi-head attention maps the $K^*$, $V^*$ vector sets from the knowledge-based encoder and the $Q$ vector set from the previous sub-layer of the decoder into an attention matrix by

$$\text{Att}_F(Q, K^*, V^*) = \text{Concat}(\text{head}_1, \ldots, \text{head}_h) W^F, \tag{9}$$

Secondly, the subsequent multi-head attention layer maps the attention matrix and the $K$, $V$ vector sets from the word-based encoder into a context representation by

$$\text{Att}_C(\text{Att}_F, K, V) = \text{Concat}(\text{head}_1, \ldots, \text{head}_h) W^C, \tag{10}$$

Thirdly, the fully connected feed-forward network is applied on the context representation by

$$\text{FFN}(\text{Att}_C) = \max\left(0, \text{Att}_C W_1^H + b_1^H\right) W_2^H + b_2^H, \tag{11}$$

Finally, a linear network layer followed by a softmax layer is applied on the outputs of the above feed-forward network layer to generate the target word sequence.

## 4. Experiment

### 4.1. Linguistic Features

In this paper, we utilized three popular linguistic features of the agglutinative language. The first was lemma, which is widely used for information retrieval. Lemmatization can make the inflected and morphological variants of the word to share representations. The second was POS tag, which can provide the syntactic role in the context. The third was morphological feature. Since different word types have distinct sets of morphological features, morphology analysis can reduce data sparseness.

### 4.2. Experimental Data

For the Turkish–English translation task, following Sennrich et al. [28], we merged the (Web Inventory of Tanscribed and Translated Talks) WIT corpus [50] and the (South-East European Times) SETimes corpus [51] as the training dataset, merged the dev2010 and tst2010 as the validation

dataset, and used the tst2011, tst2012, tst2013, tst2014 as the test datasets. For the Uyghur–Chinese translation task, we used the news data from the China Workshop on Machine Translation in 2017 (CWMT2017) as the training dataset and validation dataset and used the news data from CWMT2015 as the test dataset. Each Uyghur sentence has four Chinese references. The statistics of the training dataset on Turkish–English and Uyghur–Chinese machine translation tasks are shown in Table 1.

**Table 1.** The statistics of the training dataset for Turkish–English and Uyghur–Chinese translation.

| Language | # Sentences | # Tokens | # Word | # Lemma |
|---|---|---|---|---|
| Turkish | 355,251 | 6,712,018 | 283,858 | 96,047 |
| English | 355,251 | 8,376,414 | 110,522 | - |
| Uyghur | 330,192 | 6,043,461 | 261,918 | 128,786 |
| Chinese | 330,192 | 5,947,903 | 163,265 | - |

### 4.3. Data Preprocessing

We normalized and tokenized the experimental data. To alleviate the OOV problem, we used the byte pair encoding (BPE) method [52] to segment both the source-side and target-side words into subword units by learning separate vocabulary with 32K merge operations. In addition, we use the BPE method to segment both the Turkish lemma and the Uyghur lemma into subword units by learning separate vocabulary with 32K merge operations. Moreover, we added the "@@" token behind each non-final subword unit of the word. The segmented lemma was annotated by copying the corresponding word's other feature values to all its subword units. Thus all the linguistic annotation sequences have the same length for model training.

We utilized the Zemberek toolkit [53] with morphological disambiguation [54] to annotate the Turkish words, and we utilized the morphological analysis tool [55] to annotate the Uyghur words. Since each suffix unit in a word has one morphological feature, we concatenated all the morphological features of a word and treated it as the word's morphological tag. In particular, if a word was unknown, we used "<unk>" to annotate its POS tag and used "<null>" to annotate its morphological tag. If a word had no suffix unit, we used "<null>" to annotate its morphological tag. The training sentence examples for the Turkish–English and Uyghur–Chinese machine translation tasks are shown in Tables 2 and 3, respectively.

**Table 2.** The training sentence examples for Turkish–English machine translation task.

| Encoder | Linguistic Feature | Training Sentence Example | | | | | |
|---|---|---|---|---|---|---|---|
| word-based encoder | word | Ve bunlar sinek@@ kap@@ an@@ em@@ onlar. | | | | | |
| | lemma | ve | bu | sinek@@ | kapan | ane@@ | mon |
| knowledge-based encoder | (Part-Of-Speech) POS tag | Conj | Pron | Noun | Noun | Noun | Noun |
| | morph tag | <null> | A3pl | <null> | <null> | A3pl | A3pl |

**Table 3.** The training sentence examples for Uyghur–Chinese machine translation task.

| Encoder | Linguistic Feature | Training Sentence Example | | | | | |
|---|---|---|---|---|---|---|---|
| word-based encoder | word | لرىمز ئىككى مەدەنىيلىك @@سەنئەت خادم @@يات @@ەدەى | | | | | |
| | lemma | @@ەده | @@بىيا | @@ت | سەنئەت | خادم | ئىككى | مەدەنىيلىك |
| knowledge-based encoder | POS tag | <unk> | <unk> | <unk> | <unk> | Noun | M | Noun |
| | morph tag | <null> | <null> | <null> | <null> | NUM.PER | <null> | <null> |

### 4.4. Model Parameter Setting

We modified the Transformer model implemented in the OpenNMT-tf toolkit [56]. Both the encoder and decoder had $N = 6$ identical layers. The number of head was set to $h = 8$, and the number of the hidden units in the fully connected feed-forward network was set to 1024. Both the source-side

word and the target-side word embedding size was set to 512. We used a mini-batch size of 48 training sentences, and a maximum sentence length of 100 tokens with 0.1 label smoothing. The dropout rate in the Transformer was set to 0.1, the length penalty was set to 0.6, and the clip gradient [57] was set to 5.0. The parameters were uniformly initialized in [−0.1, 0.1]. The model was trained for 100,000 steps by using the Adam optimizer [58] with an initial learning rate of 0.0002. We reported the result of averaging the last five saved model checkpoints (saved every 5000 steps). Decoding was performed by using the beam search with a beam size of five.

We employed both the(Bilingual Evaluation Understudy) BLEU [59] and (Character n-gram F3-score) ChrF3 [60] scores to evaluate the translation performance. The vocabulary size and embedding size for the source-side Turkish and Uyghur translation tasks are shown in Table 4.

**Table 4.** The vocabulary and embedding sizes for the source-side Turkish and Uyghur translation.

| Encoder | Linguistic Feature | Vocabulary Size | | Embedding Size | |
|---|---|---|---|---|---|
| | | Turkish | Uyghur | Turkish | Uyghur |
| word-based encoder | word | 32,064 | 32,328 | 512 | 512 |
| knowledge-based encoder | lemma | 30,637 | 31,830 | 352 | 352 |
| | POS tag | 14 | 16 | 64 | 64 |
| | morph tag | 9176 | 6057 | 96 | 96 |

### 4.5. Neural Translation Models

We compared the following neural translation models with the proposed multi-source model:

- NMT baseline model: the standard Transformer model [12] without linguistic input features.
- Single-source neural model: the NMT model with linguistic features [18] that generalizes the input-embedding layer of the encoder to combine the word features and the linguistic features. All the parameter settings were the same with the multi-source neural model.

## 5. Results and Discussion

The experimental results for the Turkish–English and Uyghur–Chinese machine translation tasks are shown in Tables 5 and 6, respectively. For the Turkish–English machine translation task, from Table 5 we can see that the multi-source neural model outperformed both the NMT baseline model and the single-source neural model. It achieved the highest BLEU scores and ChrF3 scores on all the test datasets. Moreover, it achieved the highest improvements on the tst2014 dataset of 2.4 BLEU points (24.98→27.37) and 1.6 ChrF3 points (48.05→49.74). For the Uyghur–Chinese machine translation task, from Table 6 we can see that the multi-source neural model also outperformed both the NMT baseline model and single-source neural model, achieving an improvement of 0.6 BLEU points (27.60→28.21) and 0.7 ChrF3 points (36.73→37.44) on the test dataset. The experimental results show that the proposed approach is capable of effectively improving the translation performance for agglutinative languages.

**Table 5.** The experimental results on Turkish–English machine translation task.

| Neural Translation Model | BLEU Score | | | | ChrF3 Score | | | |
|---|---|---|---|---|---|---|---|---|
| | tst2011 | tst2012 | tst2013 | tst2014 | tst2011 | tst2012 | tst2013 | tst2014 |
| NMT baseline [12] | 24.18 | 25.95 | 26.60 | 24.98 | 47.18 | 48.61 | 48.65 | 48.05 |
| single-source model [18] | 24.69 | 26.65 | 27.43 | 25.98 | 47.99 | 49.26 | 49.89 | 48.87 |
| multi-source model | 25.44 | 26.75 | 28.48 | 27.37 | 48.30 | 49.80 | 50.61 | 49.74 |

**Table 6.** The experimental results on Uyghur–Chinese machine translation task.

| Neural Translation Model | BLEU Score | ChrF3 Score |
|---|---|---|
| NMT baseline [12] | 27.60 | 36.73 |
| single-source model [18] | 28.00 | 37.20 |
| multi-source model | 28.21 | 37.44 |

In addition, we found that the performance improvements of both the multi-source neural model and single-source neural model for the Uyghur-Chinese machine translation task were not very obvious. The main reason was that the morphological analysis tool for the Uyghur word annotation was not accurate enough. Since the complex morphology of Uyghur, many words cannot be effectively identified and classified, thus the tool simply annotates all the unknown words with a uniform token "<unk>" as the same with their POS tags, which increases the complexity for model training. The experimental results indicate that the annotation quality of the source-side sentence makes a difference on the translation quality of the NMT model with linguistic input features.

Figure 4 shows the loss values as a function of the time steps (saved every 100 steps) on the validation dataset in different neural translation models for the Turkish-English machine translation task. We can see that in spite of using two encoders, which leads to more model parameters and training time, the loss values of our proposed multi-source neural model is still consistent with the NMT baseline model and the single-source model. This fact indicates that our model was robust and feasible for the agglutinative language translation task. The loss value converges continuously until it achieves a lower value, then it oscillates in a small interval. Thus, we stop the NMT model training process after 10,000 steps without obvious reductions on the loss value.

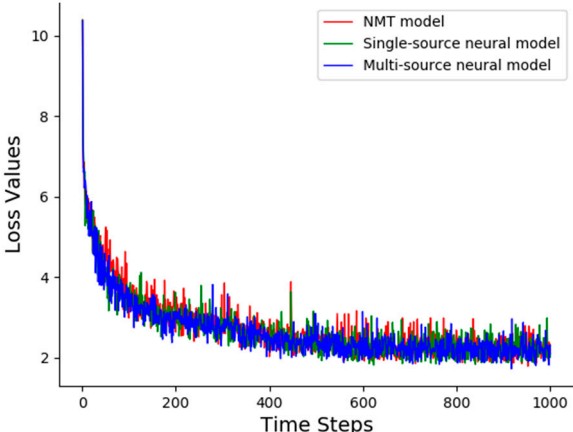

**Figure 4.** The loss values as a function of the time steps in the different neural translation models for the Turkish–English machine translation task.

To further evaluate the effect of using different linguistic features, we separately incorporated the linguistic features of lemma, POS tag, and morphological tag into the proposed multi-source neural model for comparison. The experimental results for the Turkish-English machine translation task are shown in Table 7. From the table we can find that for the test datasets of tst2011, tst2012, and tst2013, incorporating the lemma feature into the multi-source neural model achieved the highest BLEU and ChrF3 scores while for the test dataset of tst2014, incorporating the morphological feature achieved the highest BLEU and ChrF3 scores. The results indicate that different linguistic features are appropriate for different datasets. Moreover, the combination of all the linguistic features achieves the best translation quality, which demonstrates that the proposed approach enables the NMT model to better utilize the source-side linguistic features and effectively integrate them together.

**Table 7.** The experimental results of incorporating a single linguistic feature into the multi-source model.

| Input Feature | BLEU Score | | | | ChrF3 Score | | | |
|---|---|---|---|---|---|---|---|---|
| | tst2011 | tst2012 | tst2013 | tst2014 | tst2011 | tst2012 | tst2013 | tst2014 |
| word + lemma | 24.60 | 25.75 | 26.69 | 25.20 | 47.66 | 48.77 | 49.25 | 48.33 |
| word + POS tag | 24.10 | 25.54 | 26.20 | 24.98 | 47.32 | 48.66 | 48.84 | 47.98 |
| word + morph tag | 24.34 | 25.70 | 26.39 | 25.50 | 47.41 | 48.61 | 49.08 | 48.36 |

Table 8 shows the translation samples of the NMT baseline model and multi-source neural model on Turkish–English machine translation. For the first translation sample, we can observe that the NMT baseline model misunderstands the subject in the source-side sentence and simply uses a pronoun to denote the Turkish word "amerika'da" (in America). Instead, the multi-source neural model captures the above information and generates an appropriate translation result. For the second translation sample, we can observe that the NMT baseline model makes a mistake on the meaning of the Turkish word "çekiminden" (from the shooting), which leads to an inaccurate translation result. Instead, the multi-source neural model understands the semantic information of the source sentence. The above translation examples indicate that the proposed model is more sensitive to the information of the subject, location, named entity, and the word class by utilizing the source-side linguistic knowledge.

**Table 8.** The translation samples for the different NMT models for the Turkish–English translation task.

| Turkish-English Translation Samples | |
|---|---|
| source sentence 1 | Afganistan amerika'da buradan o kadar farklı görünüyor ki. |
| reference | Afghanistan looks so different from here in America. |
| NMT baseline model | It looks so different from here in Afghanistan. |
| multi-source model | Afghanistan is so different from here in America. |
| source sentence 2 | Burada bir pijama partisinde, fransız vogue çekiminden birkaç gün önce. |
| reference | Here's me at a slumber party a few days before I shot French Vogue. |
| NMT baseline model | Here's a pajama party, a few days before the French pullout. |
| multi-source model | Here I had a pajama party, a few days before French shot. |

## 6. Conclusions

In this paper, we proposed a multi-source neural model for the translation task on agglutinative language, which utilizes the source-side linguistic knowledge to enhance the representation ability of the encoder. The model employs two separate encoders to encode the word feature and the linguistic features, respectively. We first extended the input-embedding layer of an encoder to incorporate the linguistic information into NMT. Then, we used a serial combination method to hierarchically integrate the conditional information from the encoders with the outputs of the decoder, which aims to learn a high-quality context representation. The experimental results show that the proposed approach was beneficial to the translation task on the morphologically rich languages, which achieves the highest improvements of +2.4 BLEU points for the Turkish-English translation task and +0.6 BLEU points for the Uyghur-Chinese translation task. In addition, the experimental results show that the proposed multi-source neural model was capable of better exploiting the source-side linguistic knowledge and effectively integrating the linguistic features together.

In future work, we plan to utilize other combination methods to further enhance the connection between the encoder and decoder. We also plan to adjust the training parameters to find the optimal conditions for the NMT model on the low-resource and morphologically rich machine translation. Moreover, we plan to use the multi-source framework to perform transfer learning to make better generalizations on the agglutinative languages.

**Author Contributions:** Investigation, Y.P.; visualization, Y.P.; writing—original draft, Y.P.; writing—review and editing, X.L., Y.Y., and R.D. All authors have read and agreed to the published version of the manuscript.

**Funding:** This research was funded by the High-Level Talents Introduction Project of Xinjiang, grant number Y839031201; the National Natural Science Foundation of China, grant number U1703133; the National Natural Science Foundation of Xinjiang, grant number 2019BL-0006; the Open Project of Xinjiang Key Laboratory, grant number 2018D04018; and the Youth Innovation Promotion Association of the Chinese Academy of Sciences, grant number 2017472.

**Conflicts of Interest:** The authors declare no conflict of interest.

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
