# Peer review of "Multi-Source Neural Model for Machine Translation of Agglutinative Language"

_futureinternet, doi:10.3390/fi12060096_

Round 1

Reviewer 1 Report

In the reviewed paper, the authors described the ML solution for machine translation task. In general, the paper is well-written, but some parts need more justification and more analysis. My main concerns are:

1. The abstract is very chaotic. Please, rewrite it.
2. How the structure in Fig. 1 was created?
3. The figures are too small.
4. Some learning techniques should be presented.
5. Why did you use tanh() function in Eq (8)?
6. Can you present some sample vector and the results of it during some steps in your proposal?
7. The experimental section needs some work:
a) More comparison with other solutions is needed.
b) What about different architectures?
c) What about learning transfer?
d) Some more complex statistical analysis should be added

Reviewer 2 Report

The authors presented a Multi-Source Neural Model 
for synthetic non-fusional language machine translation. The primary assumption is that the model is suited for this kind of task, since the agglutination does not change the morphemes, thus keeping the model straightforward.

The authors presented the work with adequate detail. Two sets of language translations were computed which is always nice to see for applicability.

My only warning is that there are several typos that require editing for English before publication, such as:

line 12: the NMT models still struggles --> struggle

line 274: model for comparsion --> comparison

and so on...

Another minor issue is that Future Internet is not an ACL journal, meaning that the readers may not be (are probably not) linguists. Therefore, an extended tailored-for-general-readership Related Work section would be very useful for attracting interest.

  1.  

Round 2

Reviewer 1 Report

Accept.

Reviewer 2 Report

The authors addressed the comments.

The recommendation is that the paper is accepted.